# Responses to SARS-CoV-2 Vaccination in Patients with Cancer (ReCOVer Study): A Prospective Cohort Study of the Hellenic Cooperative Oncology Group

**DOI:** 10.3390/cancers13184621

**Published:** 2021-09-15

**Authors:** Helena Linardou, Nikolaos Spanakis, Georgia-Angeliki Koliou, Athina Christopoulou, Sofia Karageorgopoulou, Nephely Alevra, Anastasios Vagionas, Nikolaos Tsoukalas, Stavroula Sgourou, Elena Fountzilas, Joseph Sgouros, Evangelia Razis, Dimitra Chatzokou, Sofia Lampaki, Eleni Res, Zacharenia Saridaki, Giannis Mountzios, George Saroglou, George Fountzilas

**Affiliations:** 1Fourth Oncology Department and Comprehensive Clinical Trials Centre, Metropolitan Hospital, 18547 Athens, Greece; nephely.alevra@gmail.com (N.A.); drstevi@msn.com (S.S.); 2Department of Microbiology, Medical School, National and Kapodistrian University of Athens, 11524 Athens, Greece; nspanakis@alab.gr; 3AlfaLab, Hellenic HealthCare Group, 11524 Athens, Greece; dchatzokou@alab.gr; 4Section of Biostatistics, Hellenic Cooperative Oncology Group, Data Office, 11526 Athens, Greece; g_koliou@hecog.ondsl.gr; 5Medical Oncology Unit, S. Andrew Hospital, 26332 Patras, Greece; athinachristo@hotmail.com; 6Third Department of Medical Oncology, IASO Clinic, 15123 Athens, Greece; skarageorgopoulou@iaso.gr; 7Oncology Department, General Hospital of Kavala, 65500 Kavala, Greece; tvagionas@yahoo.com; 8Department of Oncology, 401 General Military Hospital of Athens, 11525 Athens, Greece; tsoukn@yahoo.gr; 9Second Department of Medical Oncology, Euromedica General Clinic of Thessaloniki, 54645 Thessaloniki, Greece; fountzila@oncogenome.gr; 10European University Cyprus, Nicosia 2404, Cyprus; 11Third Department of Medical Oncology, Agii Anargiri Cancer Hospital, 14564 Athens, Greece; hecogaga@otenet.gr (J.S.); nellieres@yahoo.gr (E.R.); 12Third Department of Medical Oncology, Hygeia Hospital, 15123 Athens, Greece; e.razis@hygeia.gr; 13Pulmonary Department, Lung Cancer Oncology Unit, Aristotle University of Thessaloniki, G. Papanicolaou Hospital, 57010 Thessaloniki, Greece; sofialampaki@yahoo.gr; 14Asklepieion Crete Clinic, Oncology Department, 71201 Heraklion, Greece; zeniasar@gmail.com; 15Fourth Department of Medical Oncology and Clinical Trials Unit, Henry Dunant Hospital Center, 11526 Athens, Greece; gmountzios@gmail.com; 16Medical School, National and Kapodistrian University of Athens, 11527 Athens, Greece; gs200744@otenet.gr; 17Internal Medicine Department, Metropolitan Hospital, 18547 Athens, Greece; 18Laboratory of Molecular Oncology, Hellenic Foundation for Cancer Research, Aristotle University of Thessaloniki, 54124 Thessaloniki, Greece; fountzil@auth.gr; 19Aristotle University of Thessaloniki, 54124 Thessaloniki, Greece; 20Department of Medical Oncology, German Oncology Center, Limassol 4108, Cyprus

**Keywords:** SARS-CoV-2 vaccine, cancer patient, antibody response, neutralizing IgG, anti-spike

## Abstract

**Simple Summary:**

There is limited information on the safety and efficacy of approved SARS-CoV-2 vaccines in cancer patients, as they were excluded from registration vaccine trials. We investigated the humoral immunity post SARS-CoV-2 vaccination in cancer patients compared to healthy volunteers. In this prospective cohort study, the seropositivity rate after two doses of vaccine was high in cancer patients despite active antineoplastic treatment, but their antibody titers were significantly lower than in healthy control subjects. Factors affecting immunogenicity in cancer patients, included older age, poor PS, active treatment, certain cancer types, i.e., pancreatic cancer and SCLC, male gender, and, interestingly, smoking status. Our results suggest that, given the lower immunogenicity, adjustments in vaccination strategies for more vulnerable subgroups of cancer patients may be required. Monitoring of antibody responses and elucidation of the clinical factors that influence immunity could guide future vaccination policies.

**Abstract:**

Data on the effectiveness and safety of approved SARS-CoV-2 vaccines in cancer patients are limited. This observational, prospective cohort study investigated the humoral immune response to SARS-CoV-2 vaccination in 232 cancer patients from 12 HeCOG-affiliated oncology departments compared to 100 healthcare volunteers without known active cancer. The seropositivity rate was measured 2–4 weeks after two vaccine doses, by evaluating neutralising antibodies against the SARS-CoV-2 spike protein using a commercially available immunoassay. Seropositivity was defined as ≥33.8 Binding-Antibody-Units (BAU)/mL. A total of 189 patients and 99 controls were eligible for this analysis. Among patients, 171 (90.5%) were seropositive after two vaccine doses, compared to 98% of controls (*p* = 0.015). Most seronegative patients were males (66.7%), >70-years-old (55.5%), with comorbidities (61.1%), and on active treatment (88.9%). The median antibody titers among patients were significantly lower than those of the controls (523 vs. 2050 BAU/mL; *p* < 0.001). The rate of protective titers was 54.5% in patients vs. 97% in controls (*p* < 0.001). Seropositivity rates and IgG titers in controls did not differ for any studied factor. In cancer patients, higher antibody titers were observed in never-smokers (*p* = 0.006), women (*p* = 0.022), <50-year-olds (*p* = 0.004), PS 0 (*p* = 0.029), and in breast or ovarian vs. other cancers. Adverse events were comparable to registration trials. In this cohort study, although the seropositivity rate after two vaccine doses in cancer patients seemed satisfactory, their antibody titers were significantly lower than in controls. Monitoring of responses and further elucidation of the clinical factors that affect immunity could guide adaptations of vaccine strategies for vulnerable subgroups.

## 1. Introduction

Cancer patients are at increased risk of severe Coronavirus disease 2019 (COVID-19) disease but also have significantly higher mortality rates and are, therefore, highly prioritized for vaccination against severe acute respiratory syndrome coronavirus 2 (SARS-CoV-2) [1,2,3]. In a recent meta-analysis of 52 studies and 18,650 cancer patients with COVID-19, the mortality rate was 25.6% [4]. Patients with hematological and lung malignancies, as well as those with active and/or advanced disease, have a persistently increased risk [5,6,7]. In Greece, vaccination of vulnerable groups such as elderly citizens and cancer patients of all ages was prioritized in mid-March 2021.

There are limited data on the effectiveness of vaccination in cancer patients overall, with most existing information relating to influenza vaccine [8,9,10]. Observational clinical studies suggest lower influenza mortality and morbidity in cancer patients who have been vaccinated, suggesting an effective immune response [11,12], even when undergoing systemic chemotherapy [13]. Based on extrapolation from other vaccines, the efficacy and safety of vaccination against SARS-CoV-2 in cancer patients may be similar to that in patients without cancer, although it may be generally lower in certain severely immunosuppressed subgroups. Data from clinical trials are not available because pivotal trials of the SARS-CoV-2 vaccine have excluded immunosuppressed patients. Cancer patients have a varying state of immunosuppression either due to cancer itself or treatments and complications. Therefore, cancer patients were generally excluded from the placebo-controlled randomized vaccine trials, and recommendations for them have been extrapolated from the general population. Only two vaccine trials have enrolled cancer patients, but in very small numbers: 4% cancer patients were enrolled in the Pfizer vaccine trial, and only 0.5% in the Janssen vaccine trial. However, even these patients were not analysed separately to provide information on the safety and efficacy of the vaccines [14,15,16]. A lower response to vaccination was observed in solid organ transplant recipients and, more recently, in patients with hematologic malignancies [17,18,19].

Recent reports focused on cancer patients examined antibody responses following infection with COVID-19 [19,20], while data on immune responses elicited following SARS-CoV-2 vaccination in cancer patients have been published since April 2021 [21,22,23,24,25,26,27]. From these reports, it is evident that not all cancer patients are the same. The risk of severe COVID-19 disease may be related to various cancer-related factors affecting immunocompetence [28,29], demographic or clinical parameters, leading to different levels of protection in vaccinated cancer patients. For all these reasons, international professional societies, such as ESMO, have made early calls for vaccination, but also for surveillance and monitoring of cancer patients [30].

This observational cohort study was designed to prospectively record and monitor responses of cancer patients following SARS-CoV-2 vaccination. Clinical outcomes were described, such as adverse events or COVID-19 infections after vaccination. Serological responses were monitored at three time points: before vaccination, 2–4 weeks and three months after two vaccine doses. In the present analysis, the seropositivity of cancer patients compared to control volunteers 2–4 weeks after the two doses of vaccine is presented. Possible factors that influence immunity were recorded and analysed.

## 2. Results

### 2.1. Participants’ Characteristics

A total of 232 consecutive cancer patients vaccinated against SARS-CoV-2 with the BNT162b2 (Pfizer-BioNTech, Marburg, Germany), mRNA-1273 (Moderna Biotech, Madrid, Spain S.L.) or the AZD1222 (Astra Zeneca, Leiden, the Netherlands) vaccine, and 100 hospital-personnel volunteers without active cancer at the time of vaccination who received the BNT162b2 mRNA vaccine, were enrolled. Forty-three patients and one control did not meet the criteria for the current analysis (completion of two doses and at least 2 weeks post-second dose, no documented COVID-19 infection prior to vaccination), and were excluded, leading to a total of 189 cancer patients and 99 controls (Figure 1). More than half of the patients and most of the controls were females (54% and 62.6%, respectively), with most patients being older than 60 years (64%). Volunteer healthcare workers were mostly 30–59 years old (82.8%). Three patients (1.6%) and 14.1% of the controls reported a history of autoimmune disease. Among patients, the most common cancer type was breast cancer (27%), followed by non-small cell lung cancer (NSCLC) (19.6%) and colorectal cancer (14.3%). Overall, 162 patients (85.7%) were on active antineoplastic treatment at time of vaccination with most receiving chemotherapy alone (54.9%). The baseline characteristics for the total cohort and separately for the group of patients and controls are depicted in Table 1. Most of the patients received the BNT162b2 vaccine (163 patients, 86.2%), 19 the mRNA-1273 vaccine (10.1%) and only 7 (3.7%) the AZD1222 vaccine. The median time from administration of the second vaccine dose to the date of blood sampling was 30 days (range 14–62) (32 days for patients versus 28 days for controls). In total, 120 cases (41.7%), including 98 cancer patients (51.9%) and 22 controls (22.2%), reported comorbidities at the time of first vaccine dose. Hypertension was the most reported comorbidity among both patients and controls (Appendix A).

### 2.2. Serological Outcomes

Overall, 268 cases (93.1%) had positive anti-SARS-CoV-2 anti-spike IgG antibodies: 171 patients (90.5%) and 97 (98%) controls (Fisher’s *p* = 0.015).

The median IgG titer for the patient group was 523 BAU/mL, ranging from 4.81 to 2080, and was significantly lower compared to a median of 2050 BAU/mL (range 4.81–2080) observed in the control group (Wilcoxon rank-sum *p* < 0.001, Figure 2). The rate of persons with protective titers (>488.8 BAU/mL) was 54.5% (*n* = 103) in cancer patients, as opposed to 97% in the controls (Fisher’s *p* < 0.001). The seropositivity rates and IgG titers did not differ based on gender, age, BMI, smoking status and the presence of comorbidities or autoimmune disease among controls.

Among cancer patients, a significant association was identified between IgG titers and smoking status (Kruskal–Wallis *p* = 0.017). Post-hoc analysis revealed that never smokers had significantly higher antibody titers compared to current smokers (median value 632 versus 409.5, Wilcoxon rank-sum *p* = 0.006, Figure 3A). Additionally, female patients presented with higher antibody titers compared to males (median value 665 *versus* 456, *p* = 0.022, Figure 3B). Higher antibody titers were also observed for PS 0 patients compared to those with PS 1 (median value 560 versus 253.5, *p* = 0.029, Figure 3C).

Regarding cancer type, patients with small-cell lung cancer (SCLC), colorectal and pancreatic cancer had considerably lower antibody titers as opposed to patients with breast or ovarian cancer (Appendix A). Comparison of IgG antibodies in the subgroup of patients with breast, NSCLC, SCLC, pancreatic, colorectal and ovarian cancer showed a significant difference among them, with ovarian cancer patients achieving the highest median number of antibodies among all, and those with pancreatic cancer the lowest (*p* = 0.041).

Regarding antineoplastic treatment, patients treated with endocrine therapy only and radiotherapy alone had considerably higher antibody titers; however, due to the small number of cases in these treatment groups, no formal comparisons were performed.

Further categorization of age showed that patients aged 18–49 had higher antibody rates compared to those aged 50 and over (median value 1060 versus 491.5, *p* = 0.004, Figure 3D) (Table 2). No correlation was found among patients and controls between the number of antibody titers and the number of comorbidities (Spearman rho = −0.13, *p* = 0.076 and rho = −0.03, *p* = 0.79, respectively) or the time interval between the serum sample and the date of second vaccination (rho = −0.02, *p* = 0.74 and rho = −0.13, *p* = 0.21, respectively) (Appendix A).

### 2.3. Seronegative Volunteers and Cancer Patients

Of the two seronegative controls, one had a previous renal transplant, and the other a history of autoimmune disease. More than half of the seronegative cancer patients were males (66.7%), older than 70 years (55.5%), with comorbidities (61.1%), and on active antineoplastic treatment (88.9%). Half of the seronegative patients had metastatic disease at the time of vaccination, 3 (16.7%) and had received concomitant corticosteroids, and one had an active autoimmune disease. Of note, all seronegative cancer patients were vaccinated with mRNA vaccines (BNT162b2 (*n* = 15) or mRNA-1273 (*n* = 3)).

### 2.4. Adverse Events Post Vaccination

Among healthy volunteers, the majority (74.7%, *n* = 74) reported at least one adverse event following the first vaccine dose and 68.7% (*n* = 68) following the second dose. Most of the reported events after the first dose (68/130) included topical reactions at the site of vaccination (pain, redness or local inflammation; 64.6% of controls), while fatigue/sleepiness, myalgias/arthralgias and headache were more commonly reported after the second dose (31.3%, 24.2% and 21.2%, respectively). Rarer events included tachycardia in two cases after the first and in three after the second dose, lymphadenitis and bruising in one case after the second dose and two allergic reactions requiring intravenous medication, one after the first and one after the second dose.

Among cancer patients, AEs were less commonly reported and were milder overall. More than half reported an AE after the first dose (52.4%, *n* = 99) and slightly more after the second dose (64%, *n* = 121), involving topical reaction with redness or pain in 47.1% of patients after the first and/or second dose, and fatigue (28.6%, *n* = 54), headache (24.3%, *n* = 46) or myalgias/arthralgias (9.5%, *n* = 18) after the second dose. There were no allergic reactions.

No COVID-19 infections post-vaccination were documented in either group until data cut-off for this analysis.

## 3. Discussion

In this prospective cohort study, we present real-world data on the immune responses to SARS-CoV-2 vaccination among cancer patients during the COVID-19 pandemic in Greece. We found that the immunogenicity pattern was lower in the vaccinated cancer patients than in the control subjects without cancer, but most patients were seropositive after two doses (90.5%). In 189 vaccinated cancer patients and 99 healthy volunteers, there were no documented COVID-19 infections after vaccination by the time of this analysis. In the cancer patients included in our study, antibody titers were significantly lower than in the volunteer controls despite the high seropositivity rates observed one month after vaccination, and significantly fewer cancer patients had protective antibody titers compared with the controls. We found that several factors influenced immunogenicity in the cancer patients included in this study. These primarily reflect the immunosuppressed environment of cancer patients [27] and included older age, poorer PS, active treatment, certain cancer types such as pancreatic cancer and SCLC, and interestingly, smoking status. Our study supports the widespread recommendation that cancer patients should receive a full and timely SARS-CoV-2 vaccination. Nevertheless, our results suggest that adjustments in vaccination strategies may be needed for some of the more vulnerable subgroups of cancer patients given their lower immunogenicity.

Antibody response to SARS-CoV-2 vaccines is being studied as a “correlate of protection” against infection with COVID-19 and severe disease, but the evidence is not yet clear [31,32]. The CDC does not currently recommend routine antibody testing to assess immune response to SARS-CoV-2 vaccines. However, there are recently published data showing a correlation between antibody response and disease protection in healthy volunteers and patients with malignancies, suggesting the use of post-vaccination antibody titers as a “correlate of protection” for vaccines [22,33]. Antigen-specific T cells and overall cellular immunity are thought to play a central role in the protective process against COVID-19. However, their cellular mechanisms are not yet clearly defined, and the measurements involved are far more complex than serological detection of neutralizing antibodies [34,35].

To design future strategies for the protection of vulnerable populations, such as the elderly or immunocompromised, we need to create simple ways to assess who really needs further protection. In this regard, antibody measurements could be used if we can reach methodological consensus and establish comparability between tests and laboratories. Even in high-risk populations, a serologically based vaccination strategy could provide a rational pathway for targeted booster vaccinations where they are truly needed. In some countries, a third booster vaccine dose to immunocompromised and elderly citizens has already begun and is even being considered for the general population. However, in order to keep a global vaccine equity perspective, the alternative to a universal third booster dose could be a repeat dose among high-risk populations, based on a ‘serology-monitored dosing’. This could be a strategy for rational and focused booster vaccinations among cancer patients as well. Our results support this option, as do similar data and current recommendations for cancer patients [22].

Our study revealed several factors that could influence immunity and should be considered in decision making. In vaccine pivotal trials, there were no differences in efficacy depending on age, sex, or presence of comorbidities [14]. Similarly, in our study, seropositivity rates and IgG titers did not differ by sex, age, BMI, smoking status, and the presence of comorbidities or autoimmune disease in controls. However, in cancer patients, a negative association was observed between significantly lower antibody titers and male sex, poor PS (≥1 vs. 0), and age over 50 years. Lower PS and older age likely reflect lower immunocompetence [36]. Since the beginning of the pandemic, clinical outcomes have shown that the severity and mortality of COVID-19 infection is higher in men than in women [37,38]. Recent studies suggest possible mechanisms for those gender-based variations, including differences in human ACE2 (hACE2) receptor expression, behavioural differences such as smoking or prevalence of comorbidities, and gender-based differences in immunological responses [39].

Another important finding of our study is the role of smoking in the immunological response of cancer patients. We observed a significant association between IgG titers and smoking status, with never-smokers having significantly higher antibody titers than current smokers (*p* = 0.006). Despite previous conflicting reports on smokers’ risk, it is now apparent that smoking is a predictor of COVID-19 mortality [40]. Previous reports showed that smoking has an immunosuppressive effect [41], with direct effects on T cells [42,43], affects the dendritic-cell system, and impairs host response to vaccination [44]. Recent population-based studies during the pandemic showed that smoking is a major factor associated with lower response to the SARS-CoV-2 vaccine [45]. Other recent studies showed that smokers had lower antibody titers, suggesting an impaired humoral response [46,47,48]. Our results contribute to the above evidence and show that smoking is a significant negative factor for impaired humoral immunity in vaccinated cancer patients.

In terms of cancer type, we found that patients with SCLC, colorectal and pancreatic carcinoma had significantly lower antibody titers than patients with breast or ovarian cancer, likely reflecting differences in the degree of immunosuppression due to disease, antineoplastic treatment, or gender-related differences. Because of the small number of cases in the different treatment groups, we were unable to make formal comparisons regarding specific types of treatment, but there were no numerical differences among the groups receiving immunotherapy, chemotherapy, or their combinations. In recent studies, lower or no immune responses were observed in patients with hematologic malignancies, particularly in patients undergoing anti-CD20 therapy [23,27], whereas other treatments, including immunotherapy, had no negative impact on seropositivity [27].

Nearly 10% of our cancer patients were seronegative after two doses of vaccine, similar to the recently reported percentages of 6%, 10% and 14% [22,26,27]. Consistent with other series, most seronegative patients (88.9%) in our study were on active antineoplastic treatment, in most cases with chemotherapy alone.

The safety profile of the vaccines was consistent with previous studies, suggesting that vaccination is not associated with more adverse events in cancer patients. On the contrary, fewer adverse events occurred in our cancer cohort than in the controls.

### Limitations

Regarding immunogenicity measurements, our main limitation is the lack of cellular immunity data, which would have provided a more complete picture of the vaccine’s protective mechanisms. However, our goal was to provide insight into the use of a simple, accessible, and inexpensive method to assess immunity after vaccination. In this regard, serological antibody measurements are an ideal “protective correlate”. A strength of this study is that all serological measurements were performed centrally in a reference laboratory using an approved, commercially available assay with very high clinical sensitivity and specificity. A limitation is that the upper limit of the assay is 2080 BAU/mL, so differences between groups in our study could potentially be larger.

Other limitations are that certain cancer types and treatments that could affect host immunity are poorly represented. In this regard, we were able to assess differences in specific patient subgroups but did not have sufficient power to differentiate vaccine immunogenicity in more detailed patient subgroups.

Although we did not have an age-, sex-, and comorbidity-matched control group without cancer or a control group with unvaccinated cancer patients, our results provide a clear picture of humoral immunity in real-world settings across the country. The inclusion of a large control group of hospital volunteers, who are not age-matched and thus represent a true reflection of the population, provided important comparative information. The observed differences are a strong indication that vaccination strategies against SARS-CoV-2 require tailored approaches for different populations.

## 4. Methods

### 4.1. Patients

Eligibility criteria included age > 18 years, histologically confirmed solid cancer, life expectancy greater than 3 months, active advanced or metastatic disease, or active antineoplastic treatment. Patients were enrolled in 12 HeCOG-affiliated oncology departments prior to vaccination. To be included in the present analysis, the patients needed to have been vaccinated with two doses of SARS-CoV-2 vaccine within the Hellenic National Program and be at least two and up to four weeks after the second vaccine dose. Controls were healthcare volunteers at a participating hospital (Metropolitan Hospital, Athens, Greece) who were vaccinated in January–February 2021. Patients or volunteers with a documented by positive PCR COVID-19 infection prior to study entry were excluded. In addition, active hematological malignancy or pregnancy were also excluding factors. Volunteers receiving active immunosuppressive therapy or with active cancer of any type were also excluded.

Only patients with solid tumours were included; the diagnosis of a hematological malignancy was an exclusion criterion. The cancer types included are shown in detail in Table 1. The Categories were as follows: Breast cancer, NSCLC, SCLC, Mesothelioma, Head and Neck cancer, Stomach cancer, Pancreatic cancer, Colorectal cancer, Ovarian cancer, Other Gynecological cancer, Bladder cancer, Prostate cancer, Kidney cancer, Testicular cancer, Melanoma, and Other (Neuroendocrine lung cancer, Cancer of Unknown Primary, Anal Cancer, GIST small intestine, GIST large intestine). Smoking and type of antineoplastic treatment are also shown in detail in Table 1. The categories for smoking status are as follows: Never smoker, Current smoker, Previous smoker (who stopped within the last 10 years), Not reported. Type of antineoplastic treatment was categorised as follows: Chemotherapy, Immunotherapy only, Chemo-immunotherapy combination, Targeted agent, TKI only, Other biologic therapy only, Endocrine therapy only, Radiotherapy only. These are also given in detail in Table 1. Cancer status at vaccination included: Primary recently operated, Recurrent, Metastatic, Other. Informed consent was obtained from all participants, and the study was approved by Institutional Review Board (Metropolitan Hospital, approval no. 2975, 20 January 2021) and conducted in accordance with the Declaration of Helsinki, Good Clinical Practice, and local ethical requirements. 

Clinical Trial Registration: NCT04745377

### 4.2. Study Design

This was a multi-centre, prospective, observational cohort study. The primary endpoint was the rate of seropositivity measured 2–4 weeks after two doses of vaccine in cancer patients compared with controls and expressed as antibody titers using a commercially available immunoassay. A second serologic result is planned 3 months after vaccination to determine antibody levels.

Clinical co-primary endpoint was the rate of COVID-19 infections after vaccination in cancer patients compared to controls (positive PCR tests in asymptomatic or symptomatic cases within a period of up to 12 months after vaccination).

Secondary endpoints included safety and tolerability (incidence, type, and severity of Adverse Events (AEs) after vaccination) assessed according to CTCAE v4.0.

Factors that might influence the immunity of cancer patients compared to control subjects (age, sex, smoking status, BMI and comorbidities), as well as the type of cancer and the type of antineoplastic treatment in the patient group, were correlated with the level of antibody response observed after vaccination.

### 4.3. Sample Collection and Detection of Neutralizing Antibodies to SARS-Cov-2 Spike Protein

Participants had blood samples collected before the first vaccine dose (time 1) and 2-4 weeks (time 2) after two vaccine doses. A third sample is planned 3 months after vaccination to monitor antibody levels. Serum was collected by centrifugation of whole blood in dry tubes for each patient, frozen at −20 °C and sent to a central reference laboratory, the Immunology Department of AlfaLab Molecular Biology and Genetics Center of the Hellenic Healthcare Group (HHG), Athens, Greece. A new generation chemiluminescent immunoassay (CLIA) was used for serological testing (LIAISON© SARS-CoV-2 TrimericS IgG assay), which measures the titer of neutralizing IgG antibodies against the spike protein of the SARS-CoV-2 virus. This new recombinant Trimeric Spike glycoprotein is the stabilized native form of the SARS-CoV-2 spike protein, which is thought to result in more accurate detection of IgG neutralizing antibodies. The overall clinical sensitivity of the assay is 98.7% and the clinical specificity is 99.5% [49].

The cut-off value for classifying a sample as positive was defined by the manufacturer as ≥38.8 BAU/mL and the protective titer as a value greater than 488.8 BAU/mL. The upper limit of the assay for the quantification of IgG antibodies was 2080 BAU/mL.

### 4.4. Data Recording

Questionnaires were completed on the day of sampling and contained similar questions for both groups on demographic data, personal medical history, and concomitant medications, as well as information on vaccination dates and adverse events after each vaccine dose. The cancer patients’ data included additional information from their medical records on cancer type, disease stage at the time of vaccination, type of antineoplastic treatment, and date of last systemic treatment.

### 4.5. Statistical Analysis

Baseline characteristics for patients and controls were summarized using descriptive statistics with absolute and relative frequencies for categorical variables and medians, with the respective minimum and maximum values taken into account for continuity. Comparisons of the seropositive rates between the groups of patients and controls defined by demographic and baseline characteristics were assessed via the chi-square or Fisher’s exact test, as appropriate. The Kruskal–Wallis or the Wilcoxon rank-sum test was applied to evaluate the differences in antibody titers between groups. The association of the number of antibody titers with the BMI, number of comorbidities and the time interval from the second vaccine dose to the date of blood sampling were assessed using Spearman correlations. All tests were two-sided, and significance was set at the 5% level of significance. Analysis was performed using the SAS software (SAS for Windows, version 9.4, SAS Institute Inc., Cary, NC, USA). Boxplots were created using R studio version 1.4.1106.

## 5. Conclusions

In this cohort study, the seropositivity rate after two doses of a SARS-CoV-2 vaccine was high (90.5%) in cancer patients undergoing active treatment. However, they achieved significantly lower antibody titers compared to controls. Several factors that influence immunogenicity were identified, including older age, poorer PS, active treatment, certain cancer types such as pancreatic cancer and SCLC, and smoking status. Monitoring antibody response in this population and collecting clinical data to elucidate further factors that negatively influence immunity could help in tailoring vaccination strategies for the more susceptible subgroups of cancer patients.

## Figures and Tables

**Figure 1 cancers-13-04621-f001:**
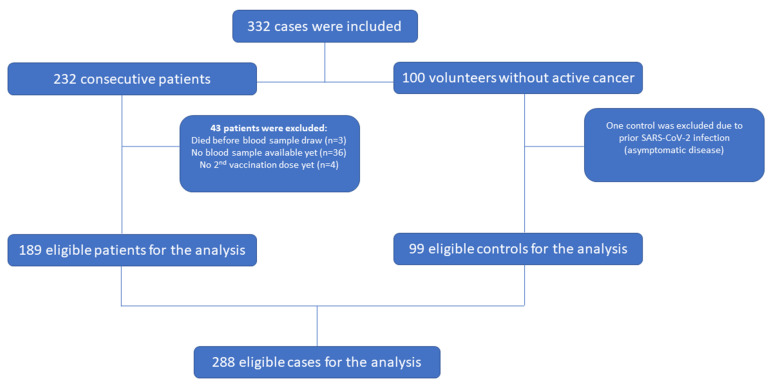
REMARK diagram.

**Figure 2 cancers-13-04621-f002:**
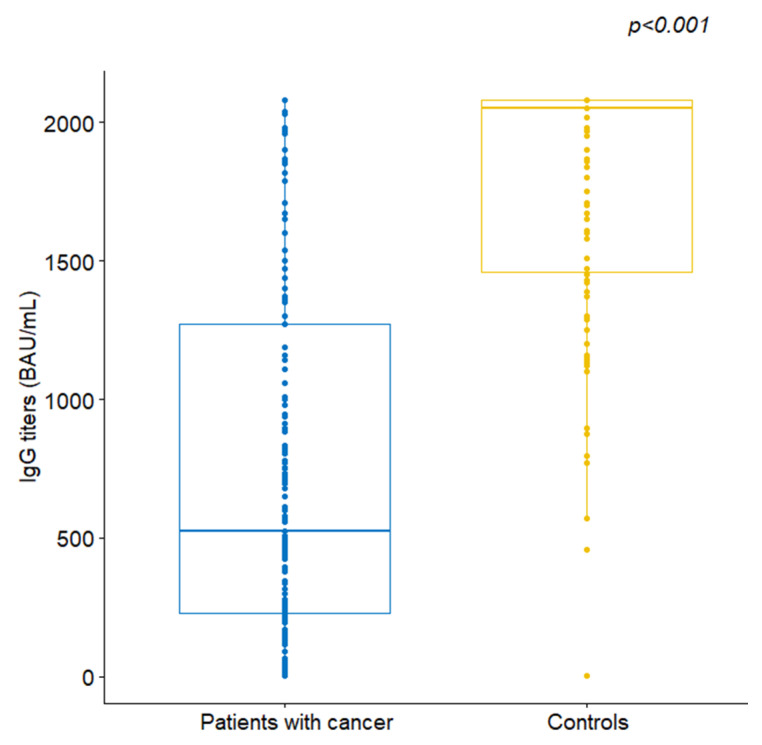
Comparison of anti-SARS-CoV-2 spike IgG antibodies between patients and controls.

**Figure 3 cancers-13-04621-f003:**
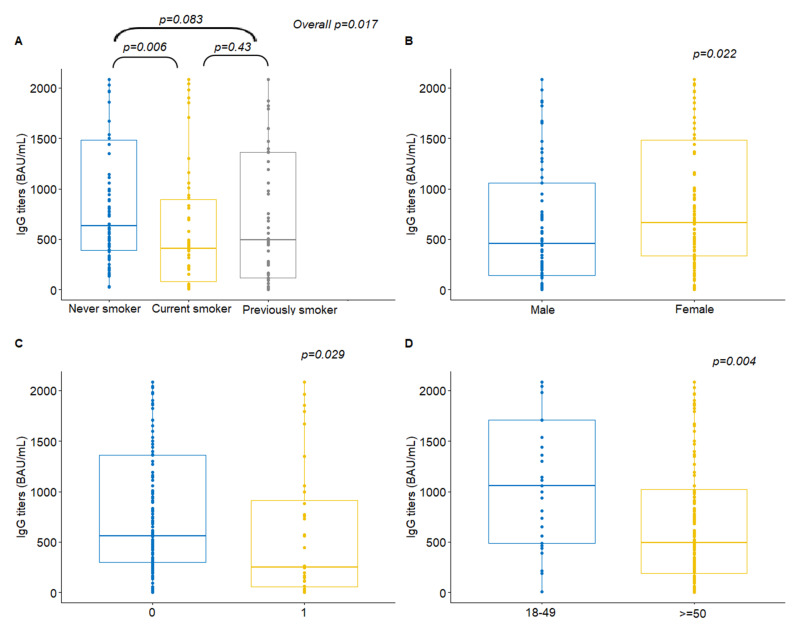
Anti-SARS-CoV-2 spike IgG antibodies by (**A**) smoking status, (**B**) gender, (**C**) performance status and (**D**) age among patients with cancer.

**Table 1 cancers-13-04621-t001:** Baseline characteristics for patients and controls.

Parameter	Total (*n* = 288)	Patients (*n* = 189)	Controls (*n* = 99)
**BMI**, Median (min, max)	25.2 (17.6, 51.3)	24.9 (17.6, 51.3)	25.4 (17.9, 45.7)
**N of comorbidities**, Median (min, max)	0.00 (0.00, 4.0)	1.00 (0.00, 4.0)	0.00 (0.00, 2.0)
	***n* (%)**	***n* (%)**	***n* (%)**
**Gender**			
Male	124 (43.1)	87 (46.0)	37 (37.4)
Female	164 (56.9)	102 (54.0)	62 (62.6)
**Age**			
18–29	5 (1.7)	1 (0.53)	4 (4.0)
30–39	28 (9.7)	3 (1.6)	25 (25.3)
40–49	57 (19.8)	25 (13.2)	32 (32.3)
50–59	64 (22.2)	39 (20.6)	25 (25.3)
60–69	60 (20.8)	51 (27.0)	9 (9.1)
70–79	56 (19.4)	52 (27.5)	4 (4.0)
80–85	15 (5.2)	15 (7.9)	0 (0.0)
>85	3 (1.0)	3 (1.6)	0 (0.0)
**BMI**			
Underweight	7 (2.4)	5 (2.6)	2 (2.0)
Normal	134 (46.5)	91 (48.1)	43 (43.4)
Obese	53 (18.4)	38 (20.1)	15 (15.2)
Overweight	94 (32.6)	55 (29.1)	39 (39.4)
**Smoking Status**			
Never smoker	124 (43.1)	82 (43.4)	42 (42.4)
Current smoker	88 (30.6)	50 (26.5)	38 (38.4)
Previous smoker—stopped within the last 10 years	71 (24.7)	53 (28.0)	18 (18.2)
Not Reported	5 (1.7)	4 (2.1)	1 (1.0)
**Comorbidities**			
No	168 (58.3)	91 (48.1)	77 (77.8)
Yes	120 (41.7)	98 (51.9)	22 (22.2)
**Vaccine administered**			
BNT162b2 (Pfizer)	262 (91.0)	163 (86.2)	99 (100.0)
mRNA-1273 (Moderna)	19 (6.6)	19 (10.1)	0 (0.0)
AZD1222 (Astra Zeneca)	7 (2.4)	7 (3.7)	0 (0.0)
**Autoimmune disease**			
No	266 (92.4)	184 (97.4)	82 (82.8)
Active autoimmune disease	17 (5.9)	3 (1.6)	14 (14.1)
Inactive autoimmune disease	4 (1.4)	1 (0.53)	3 (3.0)
Not Reported	1 (0.35)	1 (0.53)	0 (0.0)
**Cancer type (primary cancer diagnosis)**			
Breast cancer	51 (27.0)	51 (27.0)	----
NSCLC	37 (19.6)	37 (19.6)	----
SCLC	20 (10.6)	20 (10.6)	----
Mesothelioma	1 (0.53)	1 (0.53)	----
Head and Neck cancer	1 (0.53)	1 (0.53)	----
Stomach cancer	6 (3.2)	6 (3.2)	----
Pancreatic cancer	11 (5.8)	11 (5.8)	----
Colorectal cancer	27 (14.3)	27 (14.3)	----
Ovarian cancer	11 (5.8)	11 (5.8)	----
Other Gynecological cancer	1 (0.53)	1 (0.53)	----
Bladder cancer	4 (2.1)	4 (2.1)	----
Prostate cancer	4 (2.1)	4 (2.1)	----
Kidney cancer	4 (2.1)	4 (2.1)	----
Testicular cancer	1 (0.53)	1 (0.53)	----
Melanoma	3 (1.6)	3 (1.6)	----
Other *	7 (3.7)	7 (3.7)	----
**Performance Status**			
0	157 (83.1)	157 (83.1)	----
1	32 (16.9)	32 (16.9)	----
**Concomitant Corticosteroid use**			
No	179 (94.7)	179 (94.7)	----
Yes	10 (5.3)	10 (5.3)	----
**Active antineoplastic treatment**			
No	27 (14.3)	27 (14.3)	----
Yes	162 (85.7)	162 (85.7)	----
**Type of antineoplastic treatment**			
Chemotherapy	89 (54.9)	89 (54.9)	----
Immunotherapy only	32 (19.8)	32 (19.8)	----
Chemo-immunotherapy combination	12 (7.4)	12 (7.4)	----
Targeted agent, TKI only	4 (2.5)	4 (2.5)	----
Other biologic therapy only	21 (13.0)	21 (13.0)	----
Endocrine therapy only	2 (1.2)	2 (1.2)	----
Radiotherapy only	2 (1.2)	2 (1.2)	----
**Cancer status at vaccination**			
Primary recently operated	45 (23.8)	45 (23.8)	----
Recurrent	49 (25.9)	49 (25.9)	----
Metastatic	89 (47.1)	89 (47.1)	----
Other	6 (3.2)	6 (3.2)	----

*n*, number; BMI, body mass index; NSCLC, non-small cell lung cancer; SCLC, small cell lung cancer. * Neuroendocrine lung cancer (*n* = 1), Cancer of Unknown Primary (*n* = 2), Anal Cancer (*n* = 1), GIST small intestine (*n* = 1), GIST large intestine (*n* = 2).

**Table 2 cancers-13-04621-t002:** Association of antibody titers with baseline characteristics in patients and controls.

	Patients (*n* = 189)			Controls (*n* = 99)		
Parameter	*n*	IgG Median (Min, Max)	*p*-Value	*n*	IgG Median (Min, Max)	
Cancer Type (Primary Diagnosis)			0.041 *			
Breast cancer	51	698 (7.15–2080)				
Lung cancer NSCLC	37	580 (10.50–2080)				
Lung cancer SCLC	20	343.5 (18–2080)				
Mesothelioma	1	2080 (----)				
Head and Neck cancer	1	735 (----)				
Stomach cancer	6	340.50 (18.30–2080)				
Pancreatic cancer	11	238 (8.39–2080)				
Colorectal cancer	27	491.50 (35.70–2080)				
Ovarian cancer	11	939 (25.10–2080)				
Other Gynecological cancer	1	509 (----)				
Bladder cancer	4	447 (23.6–509)				
Prostate cancer	4	671.5 (4.81–883)				
Kidney cancer	4	330 (66.3–897)				
Testicular cancer	1	1980 (----)				
Melanoma	3	482 (164.00–730)				
Other	7	476 (4.81–2080)				
**Cancer status at vaccination**			0.27			
Primary recently operated	45	650 (4.81–2080)				
Recurrent	49	556 (4.81–2080)				
Metastatic	89	482 (8.07–2080)				
Other	6	729.50 (193.00–1870)				
**Active antineoplastic treatment**			0.28			
No	27	696.00 (18–2080)				
Yes	162	492.50 (4.81–2080)				
**Type of antineoplastic treatment**			----			
Chemotherapy	89	478 (4.81–2080)				
Immunotherapy only	32	469 (25.10–2080)				
Chemo-immunotherapy combination	12	536.50 (20.90–1500)				
Targeted agent, TKI only	4	378.50 (168.00–2080)				
Other biologic therapy only	21	698 (4.81–2080)				
Endocrine therapy only	2	1425.50 (771–2080)				
Radiotherapy only	2	1185 (1000–1370)				
**Concomitant corticosteroid use**			0.83			
No	179	509 (4.81–2080)				
Yes	10	817 (4.81–2080)				
**PS at vaccination**			**0.029**			
0	157	560 (4.81–2080)				
1	32	253.5 (4.81–2080)				
**Smoking status**			**0.017**			0.81
Never smoker	82	632 (25.00–2080)		42	1885 (4.81-2080)	
Current smoker	50	409.50 (8.07–2080)		38	2065 (770-2080)	
Previous smoker—stopped within the last 10 years	53	491 (4.81–2080)		18	1970 (4.81–2080)	
**Gender**			**0.022**			0.30
Male	87	456 (4.81–2080)		37	1800 (4.81–2080)	
Female	102	665 (4.81–2080)		62	2065 (770–2080)	
**Age**			----			----
18–29	1	1110 (----)		4	2025 (1700–2080)	
30–39	3	563 (215–2080)		25	2080 (770–2080)	
40–49	25	1060 (7.15–2080)		32	1905 (1100–2080)	
50–59	39	604 (20.90–2080)		25	2080 (4.81–2080)	
60–69	51	452 (4.81–2080)		9	1370 (4.81–2080)	
70–79	52	491.5 (4.81–2080)		4	1325 (4.81–2080)	
80–85	15	347 (8.07–2080)		0	----	
>85	3	730 (509–771)		0	----	
18–49	29	1060 (7.15–2080)	**0.004**	61	2080 (770–2080)	0.26
≥50	160	491.5 (4.81–2080)		38	1885 (4.81–2080)	
**Comorbidities**			0.051			0.72
No	91	650 (4.81–2080)		77	2020 (457–2080)	
Yes	98	491.5 (4.81–2080)		22	2080 (4.81–2080)	
**Hypertension**			0.76			0.077
No	120	543 (4.81–2080)		88	2080 (4.81–2080)	
Yes	69	509 (8.39–2080)		11	1510 (4.81–2080)	
**BMI**			0.80			----
Underweight	5	680 (10.5–1500)		2	1835 (1700–1970)	
Normal weight	91	509 (7.15–2080)		43	1950 (570–2080)	
Obese	38	606 (4.81–2080)		15	2080 (4.81–2080)	
Overweight	55	509 (4.81–2080)		39	1860 (4.81–2080)	

* Comparisons were performed among patients with breast, NSCLC, SCLC, pancreatic, colorectal and ovarian cancer due to the small number of cases in the rest of the patient groups.

## Data Availability

The data presented in this study are available in the article and Appendix A while further details can be obtained on request from the corresponding author.

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
