# Peer review of "Responses to SARS-CoV-2 Vaccination in Patients with Cancer (ReCOVer Study): A Prospective Cohort Study of the Hellenic Cooperative Oncology Group"

_cancers, 2021, doi:10.3390/cancers13184621_

Round 1
Reviewer 1 Report
The paper is interesting, showing a very topic study.
Few comments to improve the quality: Figure 2: the median of control samples looks to high although the samples have values distributed in a wilde range, can the authors cross check if the analysis is correct.
Table 2: I think the antibody titer of different cancer type should be converted in figure (dot plot as figure 1 and 2) because they are the most interesting data and some cancer has a lot of patients analysed which help to generate an intresting figure.
The sentences from 179 to 184 wrote the authors about the high antibody titer in patients vaccinated with moderna should be removed due to the low number of patients analysed, this result could be also due to the nature of cancer in case moderna has not be injected random.
Author Response
Thank you very much for your comments- please see here below our responses:
1. As expected, the median number of IgG antibodies in the control group of healthy individuals was considerably and statistically higher compared to the respective values in the patient group. As stated in the 2.2 section of the results, the median number of IgG titers in the control group was 2,050 BAU/mL ranging from 4.81 to 2,080 with the maximum value (i.e. 2,080) being also the upper quartile of the distribution. The value of the 25th percentile of the distribution was 1,450 and that explains the high median value of the IgG titers in the control group.
2. As suggested by the reviewer we have included a figure showing the boxplots of IgG titers by cancer type in the revised manuscript (supplemental figure 1).
3. We agree with the reviewer that the low number of patients per vaccine eliminates the power and the robustness of these results. Therefore, as per reviewer suggestion, we have excluded these sentences from the revised manuscript.
Reviewer 2 Report
The article entitled ‘Responses to SARS-CoV-2 vaccination in patients with cancer 2: a prospective cohort study of the Hellenic Cooperative Oncology Group’ which investigated the humoral immune response to SARS-CoV-2 vaccination in 232 cancer patients compared to 100 healthcare volunteers without known active cancer- in this current COVID pandemic situations It is important given the current dire COVID situation and vulnerability of cancer patients. Overall, my opinion is, it is nicely written and results would be interesting to scientific community. Few minor comments are:
In introduction,might need to explain a bit more why there is limited information on the safety and efficacy of approved SARS- CoV-2 vaccines in cancer patients.
I see they have placed the Methods at the end of the manuscript, I think the journal’s style may be;however in the methods sections they can describe bit clearly how did they categorised different cancer types and different status (such as smoking, gender and Type of antineoplastic treatment); did they find patient with hematologic malignancies?How did they select controls.
The observed significant association between IgG titers and smoking status, with never-smokers having significantly higher antibody titers than current
smokers (p=0.006). Despite previous conflicting reports on smokers' risk, it is now
apparent that smoking is a predictor of COVID-19 mortality- how did they relate this smoking status from head and neck cancer patients? Just wondering
It is important findings that in cancer patients undergoing active treatment, they achieved significantly lower antibody titers compared to controls. However, in the discussion, they did not mention how to address this lower immunity among cancer patients (eg, 3rd dose?).
Thank you.
Author Response
Response to Reviewer 2 Comments
Point 1. In introduction, might need to explain a bit more why there is limited information on the safety and efficacy of approved SARS- CoV-2 vaccines in cancer patients.
Response 1. As per reviewer’s suggestion we have expanded a bit in the introduction on the reasons of limited information on safety and efficacy of approved SARS-CoV-2 vaccines in cancer patients, as follows:
Data from clinical trials are not available because pivotal trials of the SARS-CoV-2 vaccine have excluded immunosuppressed patients. Cancer patients have a varying state of immunosuppression either due to cancer itself or the antineoplastic treatments administered. Therefore, cancer patients were generally excluded from the placebo-controlled randomized vaccine trials, and recommendations for them were based on those for the general population. Only two vaccine trials have enrolled cancer patients but in very small numbers, 4% cancer patients were enrolled in the Pfizer vaccine trial, and only 0.5% in the Janssen vaccine trial. But even these patients were not analyzed separately to provide information on safety and efficacy of the vaccines. [14-16]. A lower response to vaccination has been observed in solid organ transplant recipients and, more recently, in patients with hematologic malignancies [17,18].
Point 2. I see they have placed the Methods at the end of the manuscript, I think the journal’s style may be; however in the methods sections they can describe bit clearly how did they categorised different cancer types and different status (such as smoking, gender and Type of antineoplastic treatment); did they find patient with hematologic malignancies? How did they select controls.
Response 2. We have indeed placed the Methods at the end of the manuscript, as this was requested from the format guidelines of the journal. In order to limit the manuscript size, we had included the categorisation of different status in Table 1, with the baseline characteristics of patients and controls. However, in the revised manuscript, as per reviewer’s suggestions we added further explanatory sentences in the methods sections: only patients with solid tumours were included, the diagnosis of a hematological malignancy was an exclusion criterion. Cancer types included are shown in detail in Table 1. The Categories are as follows: Breast cancer, NSCLC, SCLC, Mesothelioma, Head & Neck cancer, Stomach cancer, Pancreatic cancer, Colorectal cancer, Ovarian cancer, Other Gynecological cancer, Bladder cancer, Prostate cancer, Kidney cancer, Testicular cancer, Melanoma, and Other (Neuroendocrine lung cancer, Cancer of Unknown Primary, Anal Cancer, GIST small intestine, GIST large intestine). Smoking and type of antineoplastic treatment are also shown in detail in Table 1. The Categories for smoking status are as follows: Never smoker, Current smoker, Previously smoker stopped within the last 10 years, Not reported. Type of antineoplastic treatment was categorised as follows: Chemotherapy, Immunotherapy only, Chemo-immunotherapy combination, Targeted agent, TKI, only, Other biologic therapy only, Endocrine therapy only, Radiotherapy only. These are also given in detail in Table 1. Cancer status at vaccination included: Primary recently operated, Recurrent, Metastatic, Other.
Point 3. The observed significant association between IgG titers and smoking status, with never-smokers having significantly higher antibody titers than current smokers (p=0.006). Despite previous conflicting reports on smokers' risk, it is now apparent that smoking is a predictor of COVID-19 mortality- how did they relate this smoking status from head and neck cancer patients? Just wondering
Response 3. We found a significant association between IgG titers and smoking status among patients with cancer underlying the negative effect of smoking on humoral immunity post SARS-CoV-2 vaccination. However, we did not perform any comparisons according to cancer type due to the low number of patients in each category that could lead to misleading results. Specifically, regarding head and neck cancer, there was only one patient with primary recently operated head and neck cancer in our study cohort who had never smoked.
Point 4. It is important findings that in cancer patients undergoing active treatment, they achieved significantly lower antibody titers compared to controls. However, in the discussion, they did not mention how to address this lower immunity among cancer patients (eg, 3rd dose?). Thank you.
Response 4. Based on the reviewer’s suggestion, we rephrased and expanded in the relevant paragraph in the discussion, as follows:
To design future strategies for the protection of vulnerable populations, such as the elderly or immunocompromised, we need to create simple ways to assess who really needs further protection. In this regard, antibody measurements could be used if we can reach methodological consensus and establish comparability between tests and laboratories. Even in high-risk populations, a serologically based vaccination strategy could provide a rational pathway for targeted booster vaccinations where they are truly needed. In some countries a third booster vaccine dose to immunocompromised and elderly citizens has already begun and is even considered for the general population. However, in order to keep a global vaccine equity perspective, the alternative to a universal third booster dose, could be a repeat dose among high-risk populations, based on a ‘serology-measurement dosing’. This could be a strategy for rational and focused booster vaccinations among cancer patients as well. Our results support this option, as do similar data and current recommendations for cancer patients [22].

Reviewer 3 Report
The manuscript titled “Responses to SARS-CoV-2 vaccination in patients with cancer (ReCOVer study): a prospective cohort study of the Hellenic Cooperative Oncology Group” by Dr. Linardou and colleagues is prospective cohort study aimed to investigate the humoral immunity post SARS-CoV-2 vaccination in a cohort of n=189 Greek cancer patients vaccinated against SARS-CoV-2 with the BNT162b2 (Pfizer), mRNA-1273 (Moderna) or the AZD1222 (Astra Zeneca) vaccine, compared to healthy volunteers, i.e., n=99 hospital-personnel volunteers without active cancer at the time of vaccination who received the BNT162b2 mRNA vaccine. The manuscript is well described and interesting data are reported about the role of covid-19 vaccination in cancer patients. Although the relatively small sample size, novel findings are presented. Conclusions are well described and well supported by the results. While recommending this study for publication following a minor revision, I have several minor observations:
Line 11 and abstract section (lines 52-81) please uniform the style
Line 79 the green marks should be removed
Line 82 it should be in bold style
Line 83 COVID-19 should be Coronavirus disease 2019 (COVID-19) when quoted for the first time
Line 85 SARS-CoV-2 should be severe acute respiratory syndrome coronavirus 2 (SARS-CoV-2) when quoted for the first time
Line lines 94-97 Thi s sentence is lacking in supporting references. A detailed information regarding the efficacy and safety of vaccination against SARS-CoV-2 is reported here (https://www.mdpi.com/1999-4915/13/9/1687; DOI 10.3390/v13091687).
Lines 118, 147, subheads should be in italic style. Also page 12
Line 120 The number of vccinated patients for each vaccine type should be detailed
Figure 2, Mentioned IgGs are anti-SARS-CoV-2 spike IgG antibodies. This information should be included in the caption and/or figure
DISCUSSION, page 12: “Several factors have been found to influence immunogenicity in cancer patients. These primarily reflect the immunosuppressed environment of cancer patients and include older age, poorer PS, active treatment, certain cancer types such as pancreatic cancer and SCLC, and interestingly, smoking status.” These sentences are lacking in supporting references. For instance (PMID: 34214473 and PMID: 34376633)
DISCUSSION, page 12: “ACE2 receptor,” it should be human ACE2 (hACE2) receptor
Author Response
Response to Reviewer 3 Comments
While recommending this study for publication following a minor revision, I have several minor observations:
Point 1. Line 11 and abstract section (lines 52-81) please uniform the style
Response 1. Style has been uniformed
Point 2. Line 79 the green marks should be removed
Response 2. Green marks removed
Point 3. Line 82 it should be in bold style
Response 3. We corrected it in bold
Point 4. Line 83 COVID-19 should be Coronavirus disease 2019 (COVID-19) when quoted for the first time
Response 4. We corrected it as suggested.
Point 5. Line 85 SARS-CoV-2 should be severe acute respiratory syndrome coronavirus 2 (SARS-CoV-2) when quoted for the first time
Response 5. We corrected it as suggested.
Point 6. Line lines 94-97 This sentence is lacking in supporting references. A detailed information regarding the efficacy and safety of vaccination against SARS-CoV-2 is reported here (https://www.mdpi.com/1999-4915/13/9/1687; DOI 10.3390/v13091687).
Response 6. The suggested reference has been added as reference 17: Rotondo, J.C.; Martini, F.; Maritati, M.; Mazziotta, C.; Di Mauro, G.; Lanzillotti, C.; Barp, N.; Gallerani, A.; Tognon, M.; Contini, C. SARS-CoV-2 Infection: New Molecular, Phylogenetic, and Pathogenetic Insights. Efficacy of Current Vaccines and the Potential Risk of Variants. Viruses. 2021; 13, 1687.
Point 7. Lines 118, 147, subheads should be in italic style. Also page 12
Response 7. We corrected as suggested.
Point 8. Line 120 The number of vaccinated patients for each vaccine type should be detailed
Response 8. In the same paragraph line 138-139 we include: Most of the patients received the BNT162b2 vaccine (163 patients, 86.2%), 19 the mRNA-1273 vaccine (10.1%) and only 7 (3.7%) the AZD1222 vaccine. Details are also given in Table 1.
Point 9. Figure 2, Mentioned IgGs are anti-SARS-CoV-2 spike IgG antibodies. This information should be included in the caption and/or figure
Response 9. We have made the suggested correction in the caption and legend of both figures in the revise manuscript.
Point 10. DISCUSSION, page 12: “Several factors have been found to influence immunogenicity in cancer patients. These primarily reflect the immunosuppressed environment of cancer patients and include older age, poorer PS, active treatment, certain cancer types such as pancreatic cancer and SCLC, and interestingly, smoking status.” These sentences are lacking in supporting references. For instance (PMID: 34214473 and PMID: 34376633)
Response 10. We rephrased the sentence as shown below, to make clear that we are referring to the factors we found in our study and we included the supporting reference (reference 27 Adeo et al. 2021) as suggested by the reviewer.
…We found that several factors influenced immunogenicity in the cancer patients included in this study. These primarily reflect the immunosuppressed environment of cancer patients [27] and …
Point 11. DISCUSSION, page 12: “ACE2 receptor,” it should be human ACE2 (hACE2) receptor
Response 11. We corrected it as suggested.
